

# Constructing non-Abelian quantum spin liquids using combinatorial gauge symmetry

**Dmitry Green[1,2]⋆ and Claudio Chamon[1]†**

**1** Physics Department, Boston University, Boston, MA, 02215, USA
**2** AppliedTQC.com, ResearchPULSE LLC, New York, NY 10065, USA

⋆ dmitry.green@aya.yale.edu , † chamon@bu.edu

## Abstract

We construct Hamiltonians with only 1- and 2-body interactions that exhibit an *exact* non-Abelian gauge symmetry (specifically, combinatorial gauge symmetry). Our spin Hamiltonian realizes the quantum double associated to the group of quaternions. It contains only ferromagnetic and anti-ferromagnetic $ZZ$ interactions, plus longitudinal and transverse fields, and therefore is an explicit example of a spin Hamiltonian with no sign problem that realizes a non-Abelian topological phase. In addition to the spin model, we propose a superconducting quantum circuit version with the same symmetry.

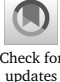

# 1 Introduction

Non-Abelian topological states are some of the most remarkable forms of quantum matter. The exchange of quasiparticle excitations in these systems is characterized by non-commuting unitary transformations in a space of degenerate many-body states, i.e., these quasiparticles have non-Abelian braiding statistics [1, 2]. Non-Abelian states are theoretically predicted to describe certain fractional quantum Hall (FQH) states [3–6]. Kitaev's honeycomb spin-liquid model [7] is another example; it displays a non-Abelian phase in the presence of a magnetic field, with excitations that have Ising-anyon statistics. A more general class of systems that realize non-Abelian topological states of matter is that of Kitaev's exactly solvable quantum double models [8], in which the specific state is determined by the choice of the non-Abelian group in which the link (or gauge) degrees of freedom take their values.

An obstacle to realize the quantum double models in experimental systems is that they are written in terms multi-body interactions among degrees of freedom expressed as group elements, not physical degrees of freedom, such as spins or charges. To implement the quantum doubles experimentally would require designing parent Hamiltonians with 1- and 2-body interactions. Notable efforts along these lines have been made in Refs. [9, 10] and [11]. The local gauge symmetries in quantum double realizations of Refs. [9, 10] are emergent, being active only in the low energy sector of the theory (hence perturbative). On the other hand, the local gauge symmetries are exact in the case of Ref. [11], but it is not clear that the Hamiltonian is physically realizable like in Ref. [9], where a physical implementation is proposed using arrays of Josephson junctions. The goal of this paper is to develop a framework that fills in the gaps in both of these approaches: we design a physical Hamiltonian with exact local non-Abelian gauge symmetries, using only 1- and 2-body interactions that could be implemented in physical systems, such as superconducting quantum circuits.

The program hinges on extending combinatorial gauge symmetry [12] (see Ref. [13] for an in depth introduction to the symmetry principle for Abelian theories that is accompanied by step-by-step constructions of examples) to a non-Abelian theory. The gauge symmetries are built into the *microscopic* Hamiltonian, and hence are *exact*, as opposed to emerging only in a low energy limit. That the gauge symmetry is exact in realistic Hamiltonians expands the range of parameters for which the topological phase may be stable, thus providing a way to escape limits on the sizes of the attainable energy gaps. Moreover, the model has ferromagnetic and anti-ferromagnetic $ZZ$ interactions, plus longitudinal and transverse fields. Therefore, the spin model is an *explicit* realization of a spin Hamiltonian without a sign problem that realizes a non-Abelian topological phase.

We focus on the quantum double with link (or leg) variables taking values within the quaternion group, $Q_8$, on a honeycomb lattice. We represent the 8 quaternion variables ($\pm 1, \pm i, \pm j$, and $\pm k$) with spin-1/2 degrees of freedom. We shall utilize 4 "gauge" spins in each link of the honeycomb lattice, thereby defining a 16-dimensional Hilbert space that we split into two sets, of even and odd parity states, and use the 8 even parity states to represent the 8 quaternions. The construction utilizes "matter" spins on the links to split the even and odd parity states and on the sites to enforce that the three quaternion variables multiply to the identity ("zero flux" condition).

Finally, we present a superconducting quantum circuit with the same non-Abelian combinatorial gauge symmetry. In the limit where the superconducting wires are small, and voltage biases are tuned so that two nearly degenerate charge states are favored in each wire, the system becomes a non-Abelian generalization of the WXY model introduced in Ref. [14]. In this case, the remaining energy scale in the problem is the Josephson coupling, and if the system (with the combinatorial gauge symmetry) is gapped, the non-perturbative gap must be necessarily on the order of this scale.

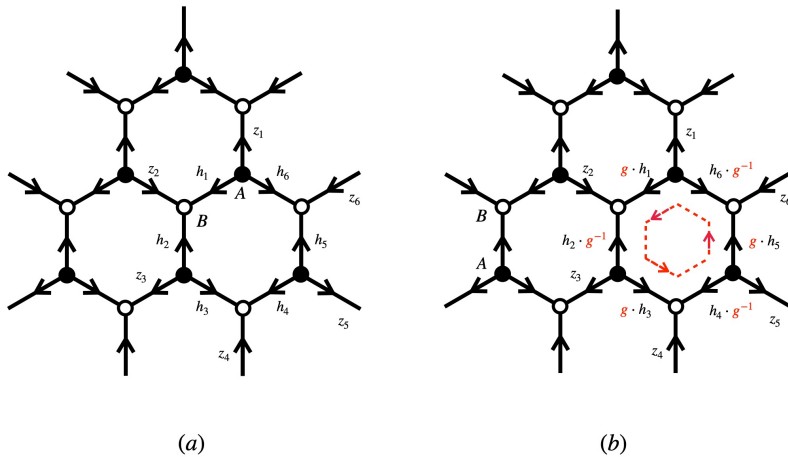

$$(a) \qquad\qquad (b)$$

Figure 1: Representation of a finite non-Abelian gauge theory on the hexagonal lattice. (a) Each link is oriented and a degree of freedom, taking values on a group $G$, lives on it. Examples of link degrees of freedom are those labeled by $z_i \in G$ and $h_i \in G$. We have chosen link orientations such that sublattices $A$ and $B$ have all arrows pointing out and in, respectively. In the ground state the zero "flux" on each vertex is equivalent to either clockwise (sublattice $A$) or counterclockwise (sublattice $B$) multiplication of the group elements on the associated legs to the identity. (In the ground state, $z_1 \cdot h_6 \cdot h_1 = 1$ and $z_2 \cdot h_2 \cdot h_1 = 1$ on $A$ and $B$, respectively.) The order of multiplication is important as the group is non-Abelian. (b) Local gauge symmetry associated with a group element $g$ is shown in red. The plaquette is denoted by an oriented path in red. Each time a link is traversed in the same direction as its orientation the group element on the leg is multiplied by $g$ from the left. Conversely, traversing along a leg against its orientation multiplies the group element on the leg by $g^{-1}$ from the right. The flux is preserved on each vertex around the plaquette. Note again that the order matters because the group is non-Abelian.

## 2 Preliminaries and a roadmap

Before diving into details, we think that it will aid the presentation to very briefly summarize the basic elements of the quantum double [8] and how we will use them. For an even more general introduction to discrete gauge theories see Ref. [15] (this reference also includes an explicit construction of excitations and their fusion rules for the quaternion group).

The Hilbert space for a finite gauge theory on a lattice is spanned by states $|z\rangle$ on each link, where $z$ is an element in a group $G$. Each link also has an (arbitrary) orientation which, for a non-Abelian group, is necessary to define how operators act within the Hilbert space. See Fig. 1.

Conventionally, the notion of flux is defined by a product of the group elements around plaquettes. In our approach we find it convenient to use the dual lattice, instead, where flux is defined on the vertices by the product of group elements on the legs around the vertex. We refer the reader to Kitaev's work on the quantum double for details.

The gauge symmetry can be thought of as the insertion of any group element $g$ into any plaquette. In this operation the states on the links surrounding the plaquette are multiplied in a specific order such that the flux at each vertex is preserved. Since the group elements do not commute, some link states are multiplied from the left and some from the right (see Fig. 1). The Hamiltonian is invariant under any such plaquette transformation.

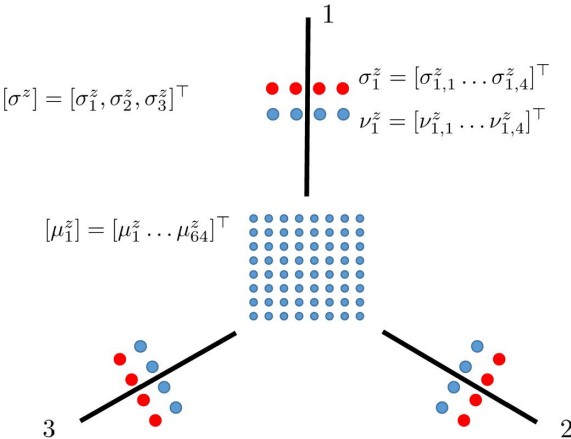

Figure 2: Spin construction for the system with non-Abelian gauge symmetry. Spin-1/2 degrees of freedom are placed on links and on the vertices of the lattice (one vertex and three links or legs are shown). Gauge spins are depicted in red, and matter spins, placed both at the links and at the vertex, are depicted in blue. There are two steps in the construction. First, the Hilbert space of quaternions on each leg $\alpha = 1, \ldots, 3$ is implemented by a set of gauge spins, $\sigma_\alpha^z$, and a companion set of four matter spins, $\nu_\alpha^z$, that are coupled. Each of $\sigma_\alpha^z$ and $\nu_\alpha^z$ are 4-spinors and the specific form of the coupling between them serves to implement the quaternion group at each site. The combined 12-spinor of gauge spins is denoted by $[\sigma^z]$. Second, all 12 gauge spins are coupled to an additional set of 64 matter spins $\mu_n^z$ ($n = 1, \ldots, 64$) that are at the center of the vertex. The coupling is all-to-all, i.e., there are $12 \times 64$ couplings between $[\sigma^z]$ and $[\mu^z]$. The specific form of this coupling will serve to implement the local quaternion gauge symmetry. Note that the matter spins $\nu^z$ and $\mu^z$ are not coupled and that the gauge spins are shared by neighboring vertices in the sense that they couple to matter spins $\mu^z$ residing at the center of neighboring vertices.

The core challenge in realizing such a model is two-fold. First, what are the physical elements that can be used to represent the abstract group elements $g$? Second, how do we construct a Hamiltonian that is invariant under these abstract operations where vertex and plaquette terms are products of multi-leg terms?

We are able to address both challenges for the quaternion group on a hexagonal lattice. For this case, we find a representation of both the group elements and the Hamiltonian by ordinary spin-1/2 states that are coupled by simple Ising interactions.

At the heart of our approach is combinatorial gauge symmetry. Schematically, suppose we have an Ising Hamiltonian that couples two sets of spins $\mu_n^z$ and $\sigma_i^z$. The $\sigma$ are what we call gauge spins and $\mu$ are matter spins. All the gauge symmetries that we want to emulate are embodied by the gauge spins, while the matter spins serve to enable the symmetry via permutations of states within the enlarged Hilbert space.

Consider the general Hamiltonian at a given site $s$ (details do not matter for the purposes of the roadmap): $H_s = \sum_{ni} \mu_n^z W_{ni} \sigma_i^z$ with $n = 1, \ldots, p$ and $i = 1, \ldots, q$. Now suppose we want to represent a given group operator $g$. We construct a $q \times q$ representation matrix $R(g)$ that acts on the set of $\sigma_i^z$ two legs at a time. For a suitably chosen set of interactions $W_{ni}$ we require that there is a companion $p \times p$ permutation matrix $L(g)$ that acts on the matter spins $\mu_n^z$ such that the Hamiltonian is invariant. In other words we demand the automorphism $L^{-1}(g) W R(g) = W$ for all $g$ on all sites $s$ and hence gauge invariance. A key feature is that both $L$ and $R$ transformations must be monomial matrices in order to preserve the spin com-

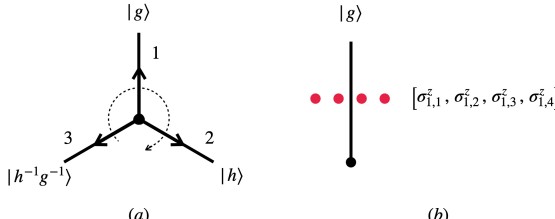

Figure 3: (a) Dual lattice version of Kitaev's quantum double. The ground states are defined by configurations in which the product of the allowed group elements clockwise (orientation matters) is $g_1 g_2 g_3 = 1$, where $g_1 = g$, $g_2 = h$ and $g_3 = (gh)^{-1} = h^{-1}g^{-1}$. (b) The quaternion group element on each leg is represented by a 4-vector of the eigenvalues ($\pm$) of four spins. The figure shows leg 1 as an example, with the red dots representing the four gauge spins (matter spins are not shown for simplicity).

mutation relations. In this way we are able to represent all operations of the group elements $g$ on the gauge spins by absorbing them into permutations of matter spins. This symmetry is exact by construction. In effect we have reduced the problem to that of finding an appropriate $W$ and representations of $R$ and $L$ in terms of spins.

Once we have constructed the appropriate representation we will need to ensure that the ground state manifold is in fact composed only of the states that satisfy the "zero flux" condition that the group elements on a star multiply to the identity. We will find that it is necessary to add a longitudinal field on some of the matter spins (respecting the combinatorial symmetry), but this is a straightforward interaction from a physical point of view.

The geometry of our solution is shown in Fig. 2; we describe it in detail below.

## 3 Spin representation of quaternions and left/right representations

Consider quaternion degrees of freedom placed on the links of a honeycomb lattice. Let us focus on the vertices of the lattice, as depicted in Fig. 3(a), and construct interactions that favor the configurations in which the three link variables $g_1, g_2, g_3 \in Q_8$ multiply (clockwise) to the group identity: $g_1 g_2 g_3 = 1$. (Note that we focus on the 3-legged stars of the honeycomb lattice, as opposed of the 3-sided plaquettes of a triangular lattice as in Kitaev's original formulation; these two systems are equivalent, just formulated in dual lattices.) We represent the quaternion elements using 4 spin-1/2 variables, as illustrated in Fig. 3(b).

The Hilbert space for the link variable is spanned by the states $|g\rangle$, with $g \in Q_8$. The formulation of the quantum double of Ref. [8] defines operators $L_+^h |g\rangle \equiv |hg\rangle$ and $L_-^h |g\rangle \equiv |gh^{-1}\rangle$ that multiply the group element inside the ket on its left or right. In representing the elements of the quaternion group in terms of spins, we shall define monomial matrices $\ell(h)$ and $r(h)$ that will have similar effects on the spin states (see Ref. [16] for an exercise on writing left/right representations of $Q_8$; we introduce and use different matrices here that are suitable for basis vectors that contain only $\pm 1$ elements that represent spins rather than 0's and 1's.)

Let us associate the following 4-vectors to the group elements of $Q_8$:

$$
\begin{aligned}
v(+1) &= \left[+++ +\right], & v(-1) &= \left[----\right], \\
v(+i) &= \left[+-+-\right], & v(-i) &= \left[-+-+\right], \\
v(+j) &= \left[++--\right], & v(-j) &= \left[--++\right], \\
v(+k) &= \left[-++-\right], & v(-k) &= \left[+--+\right].
\end{aligned}
\tag{1}
$$

The $\pm$ stand for $\pm 1$. We will use spins to represent each entry in the vectors $v$, in which case the $\pm$ should be thought of as representations of the two eigenstates in the $\sigma^z$ basis: $+ \equiv (1\ 0)$ and $- \equiv (0\ 1)$. Notice that all $v$ have even parity.

One can represent the action of left and right multiplication by group elements through matrices $\ell(g)$ and $r(g)$ such that

$$
\begin{aligned}
v(g)\,\ell(h) &= v(hg), \\
v(g)\,r(h) &= v(gh).
\end{aligned}
\tag{2}
$$

The $4 \times 4$ monomial matrix representations for these left/right operators are

$$
r(\pm 1) =
\begin{bmatrix}
\pm & 0 & 0 & 0 \\
0 & \pm & 0 & 0 \\
0 & 0 & \pm & 0 \\
0 & 0 & 0 & \pm
\end{bmatrix}, \quad
r(\pm i) =
\begin{bmatrix}
0 & \mp & 0 & 0 \\
\pm & 0 & 0 & 0 \\
0 & 0 & 0 & \mp \\
0 & 0 & \pm & 0
\end{bmatrix},
$$

$$
r(\pm j) =
\begin{bmatrix}
0 & 0 & 0 & \mp \\
0 & 0 & \mp & 0 \\
0 & \pm & 0 & 0 \\
\pm & 0 & 0 & 0
\end{bmatrix}, \quad
r(\pm k) =
\begin{bmatrix}
0 & 0 & \pm & 0 \\
0 & 0 & 0 & \mp \\
\mp & 0 & 0 & 0 \\
0 & \pm & 0 & 0
\end{bmatrix},
\tag{3a}
$$

and

$$
\ell(\pm 1) =
\begin{bmatrix}
\pm & 0 & 0 & 0 \\
0 & \pm & 0 & 0 \\
0 & 0 & \pm & 0 \\
0 & 0 & 0 & \pm
\end{bmatrix}, \quad
\ell(\pm i) =
\begin{bmatrix}
0 & 0 & 0 & \mp \\
0 & 0 & \pm & 0 \\
0 & \mp & 0 & 0 \\
\pm & 0 & 0 & 0
\end{bmatrix},
$$

$$
\ell(\pm j) =
\begin{bmatrix}
0 & 0 & \mp & 0 \\
0 & 0 & 0 & \mp \\
\pm & 0 & 0 & 0 \\
0 & \pm & 0 & 0
\end{bmatrix}, \quad
\ell(\pm k) =
\begin{bmatrix}
0 & \pm & 0 & 0 \\
\mp & 0 & 0 & 0 \\
0 & 0 & 0 & \mp \\
0 & 0 & \pm & 0
\end{bmatrix}.
\tag{3b}
$$

Note that both $r$ and $\ell$ preserve the parity of each $v$ because they are monomial matrices with an even number of $-1$'s. Also note that $\ell(h^{-1}) = \ell^\top(h)$ and $r(h^{-1}) = r^\top(h)$. In terms of their action on the spin degrees of freedom, the $+$ and $-$ signs inside these matrices should be interpreted as $\mathbb{1}_2$ and $\sigma^x$ operators acting on the underlying spins; the latter flips a spin and the former is the $2 \times 2$ identity operator.

One can also construct a convenient $4 \times 4$ matrix $w$ that implements inversion of a group element $h$:

$$
v(h)\,w = v(h^{-1}).
\tag{4}
$$

$w$ turns out to be the Hadamard matrix

$$
w = \frac{1}{2}
\begin{bmatrix}
- & + & + & + \\
+ & - & + & + \\
+ & + & - & + \\
+ & + & + & -
\end{bmatrix}.
\tag{5}
$$

This matrix ties the left/right representations together:

$$w\, r(h)\, w = \ell(h^{-1}) \quad \text{and} \quad w\, \ell(h)\, w = r(h^{-1})\,. \tag{6}$$

Crucially, it follows that $w$ is invariant under left/right monomial transformations,

$$\ell(h)\, w\, r(h) = w \quad \text{and} \quad r(h)\, w\, \ell(h) = w\,. \tag{7}$$

This is a key relation, and it is one of two automorphisms that we will use in our construction of the Hamiltonian (per Sec. 2).

## 4 Building the quaternion quantum double Hamiltonian

We construct the Hamiltonian for the quaternion quantum double in two steps. The first step is to construct terms to separate 16 states associated with 4 spin-1/2 degrees of freedom, in each link, into two sets of 8 states each. One set of states – those with even parity – will correspond to the representation of the elements of $Q_8$ as described above; the other set of 8 states – those with odd parity – will be pushed up in energy, as discussed in Sec. 4.1 below.

The second step of the construction is to design in Sec. 4.2 a Hamiltonian that is, as shown in Sec. 4.3, invariant under transformations associated to left/right multiplication of the link variables on a vertex by group elements that leave vertices where the product $g_1 g_2 g_3 = 1$ invariant, as illustrated in Fig. 4. We note that the quaternion combinatorial gauge symmetry itself, as we shall see, is independent of the even-odd splitting, so the construction is non-perturbative, as will become explicit below.

### 4.1 Hamiltonian on the links

On each of the links of the honeycomb lattice, labeled $\alpha$, we place four spins that we collect into the 4-spinor $\sigma_\alpha^z$:

$$\sigma_\alpha^z = \begin{bmatrix} \sigma_{\alpha,1}^z \\ \sigma_{\alpha,2}^z \\ \sigma_{\alpha,3}^z \\ \sigma_{\alpha,4}^z \end{bmatrix}\,. \tag{8}$$

In the $z$-basis, each $\sigma_\alpha^z$ will form the basis representation of the group elements as in Eq.(1) and each transforms under $r$ and $\ell$ as in in Eq.(2). To ensure that we will be able to project to the even-parity subspace we introduce four additional spins, represented by the 4-spinor $\nu_\alpha^z$,

$$\nu_\alpha^z = \begin{bmatrix} \nu_{\alpha,1}^z \\ \nu_{\alpha,2}^z \\ \nu_{\alpha,3}^z \\ \nu_{\alpha,4}^z \end{bmatrix}\,, \tag{9}$$

which are coupled to the $\sigma_\alpha^z$ via the Ising interactions

$$H_{\text{leg},\alpha}^{\text{Ising}} = -K\, \nu_\alpha^{z\,\top}\, w\, \sigma_\alpha^z\,, \tag{10}$$

with couplings proportional to the Hadamard matrix in Eq.(5). These couplings are the same as those described in Ref. [12], and they favor the even-parity subspace ($\sigma_{\alpha,1}^z \sigma_{\alpha,2}^z \sigma_{\alpha,3}^z \sigma_{\alpha,4}^z = 1$). These couplings are invariant under monomial transformations. For example if $\sigma_\alpha^z \to r(g)\sigma_\alpha^z$ then $\nu_\alpha^{z\,\top} \to \nu_\alpha^{z\,\top}\ell(g)$ preserves the Hamiltonian because of the automorphism of $w$ in Eq.(7).

To the Ising couplings in Eq. (10) we can add uniform transverse fields, which are invariant under the transformations $r$ and $\ell$, because they are invariant under both flips of the $z$-components and permutations of matter spins. The general Hamiltonian on each leg is then:

$$H_{\text{leg},\alpha} = -K\, \nu_\alpha^{\text{z}\,\top}\, w\, \sigma_\alpha^{\text{z}} - \Gamma_\sigma \sum_i \sigma_{\alpha,i}^{\text{x}} - \Gamma_\nu \sum_i \nu_{\alpha,i}^{\text{x}}\,. \tag{11}$$

## 4.2 Hamiltonian on the vertex

In the second step of the construction we recursively collect the three legs meeting at a vertex, introduce matter spins at the vertex, and construct a gauge-matter coupling matrix $W$. $W$ will be invariant under non-Abelian gauge transformations and will also favor the configurations in which the product of the quaternions on those three legs equals the identity.

We collect the three 4-spinors on each leg into a 12-spinor $[\,\sigma^{\text{z}}\,]$,

$$[\,\sigma^{\text{z}}\,] = \begin{bmatrix} \sigma_1^{\text{z}} \\ \sigma_2^{\text{z}} \\ \sigma_3^{\text{z}} \end{bmatrix}. \tag{12}$$

We also define 64-spinor of matter spins

$$[\,\mu^{\text{z}}\,] = \begin{bmatrix} \mu_{f_1,h_1}^{\text{z}} \\ \vdots \\ \mu_{f_{64},h_{64}}^{\text{z}} \end{bmatrix}, \tag{13}$$

where the 64 states correspond to the choices of 64 triplets of group elements that multiply to 1, parametrized as $f_i$, $h_i$, and $(h_i f_i)^{-1}$, $i = 1,\ldots,64$.

Now we couple the gauge and matter spins as follows

$$H_{\text{vertex}}^{\text{Ising}} = -J\,[\,\mu^{\text{z}}\,]^\top\, W\,[\,\sigma^{\text{z}}\,]\,. \tag{14}$$

The the $64 \times 12$ matrix $W$ is the interaction matrix defined by

$$W = \frac{1}{4} \begin{bmatrix} \nu(f_1) & \nu(h_1) & \nu((f_1 h_1)^{-1}) \\ \nu(f_2) & \nu(h_2) & \nu((f_2 h_2)^{-1}) \\ \vdots & \vdots & \vdots \\ \nu(f_{64}) & \nu(h_{64}) & \nu((f_{64} h_{64})^{-1}) \end{bmatrix}. \tag{15}$$

Each $\nu$ is exactly as defined by Eq. (1) and therefore the interactions are $\pm J$, i.e., ferromagnetic or anti-ferromagnetic. This interaction matrix enumerates all configurations that satisfy the flux relation on the vertex in Fig. 3. In each row we have that the product of the quaternion group elements represented by the three $\nu$'s in each row *from left to right* is 1 (order matters). The number of rows exhausts the list of all possible leg configurations, matching the number of matter spins.

We can also apply transverse and longitudinal fields to the matter spins, while preserving the combinatorial symmetry, because they are invariant under permutations of the matter spins at each vertex. The more general Hamiltonian is then:

$$H_{\text{vertex}} = -J\,[\,\mu^{\text{z}}\,]^\top\, W\,[\,\sigma^{\text{z}}\,] - \Gamma_\mu \sum_i \mu_{f_i,h_i}^{\text{x}} - H_\mu \sum_i \mu_{f_i,h_i}^{\text{z}}\,. \tag{16}$$

On the full lattice, the Hamiltonian is the sum of the leg terms in Eq. (11) and vertex terms in Eq. (16). We denote the legs emanating from each vertex $s$ by $\alpha(s)$ so that the full Hamiltonian is:

$$H = \sum_s H_{\text{leg},\alpha(s)} + H_{\text{vertex},s}\,. \tag{17}$$

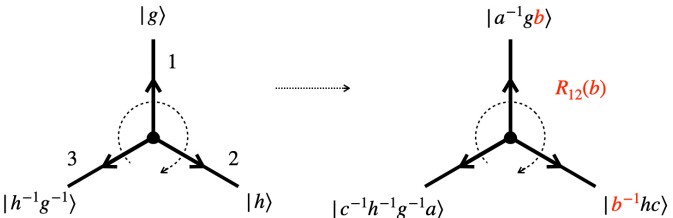

Figure 4: Most general transformation of the hexagonal vertex that preserves the flux, where $a$, $b$ and $c$ are group elements. $R_{12}(b)$ is highlighted in red as an example.

## 4.3 Non-Abelian combinatorial gauge symmetry

Now we show that each term in the full lattice Hamiltonian Eq. (17) is invariant under local non-Abelian transformations. Let us start with the $ZZ$-term of the vertex Hamiltonian Eq. (16), and consider transformations as depicted in Fig. 4. Consider three $12 \times 12$ block diagonal matrices, acting on the 12-spinor $[\sigma^z]$, for group elements $a$, $b$, and $c$:

$$R_{31}(a) = \begin{bmatrix} \ell(a^{-1}) & 0 & 0 \\ 0 & \mathbb{1}_4 & 0 \\ 0 & 0 & r(a) \end{bmatrix}, \tag{18a}$$

$$R_{12}(b) = \begin{bmatrix} r(b) & 0 & 0 \\ 0 & \ell(b^{-1}) & 0 \\ 0 & 0 & \mathbb{1}_4 \end{bmatrix}, \tag{18b}$$

$$R_{23}(c) = \begin{bmatrix} \mathbb{1}_4 & 0 & 0 \\ 0 & r(c) & 0 \\ 0 & 0 & \ell(c^{-1}) \end{bmatrix}, \tag{18c}$$

where $\mathbb{1}_4$ is the $4 \times 4$ identity matrix and the matrices $\ell(h)$ and $r(h)$ are the $4 \times 4$ left/right representations of quaternions. The matrices in Eq. (18) act on two of the three legs on the vertex, one by multiplying by a group element on the left, and one by the (inverse) group element on the right. Each of these transformations preserve the zero flux condition on the vertex (i.e., that the three group elements multiply to the identity). Each $R_{ij}$ obeys the quaternion algebra $R_{ij}(g)R_{ij}(h) = R_{ij}(gh)$ for all $i,j$, which follows from $r(g)r(h) = r(gh)$ and $\ell(g)\ell(h) = \ell(hg)$. When one of the three legs is different, the $R$'s commute, which follows from the commutation of the $r$ and $\ell$ block matrices. The latter property ensures that we will be able to insert charges locally because the two operations of inserting a group element into one hexagon and another group element into a neighboring hexagon in Fig.1 will commute, as they correspond to multiplication on the left and right.

For example, the transformation $R_{12}(b)$ is pictured in Fig. 4b. Generically $R_{12}(b)$ has the effect of mapping group elements $f_i$ and $h_i$ by

$$\begin{aligned} f_i &\to f_i\, b\,, \\ h_i &\to b^{-1} h_i\,, \end{aligned} \tag{19}$$

for all lines $i = 1, \ldots, 64$ of the matrix $W$ in Eq. (15). Explicitly:

$$
W R_{12}(b) = \begin{bmatrix}
v(f_1)\, r(b) & v(h_1)\, \ell(b^{-1}) & v((f_1 h_1)^{-1}) \\
v(f_2)\, r(b) & v(h_2)\, \ell(b^{-1}) & v((f_2 h_2)^{-1}) \\
\vdots & \vdots & \vdots \\
v(f_{64})\, r(b) & v(h_{64})\, \ell(b^{-1}) & v((f_{64} h_{64})^{-1})
\end{bmatrix}
$$

$$
\equiv \begin{bmatrix}
v(f_1 b) & v(b^{-1} h_1) & v((f_1 h_1)^{-1}) \\
v(f_2 b) & v(b^{-1} h_2) & v((f_2 h_2)^{-1}) \\
\vdots & \vdots & \vdots \\
v(f_N b) & v(b^{-1} h_N) & v((f_N h_N)^{-1})
\end{bmatrix}, \tag{20}
$$

where we used Eq.(2). Note that $W R_{12}(b)$ contains all the same 64 lines of $W$, but permuted, i.e., we can write:

$$
W R_{12}(b) = L_{12}(b)\, W,
$$

where $L_{12}(b)$ is a $64 \times 64$ permutation matrix. In general, for any matrix $R$ that is a product of the $R_{ij}$ in Eq.(18) there is a corresponding permutation matrix $L$ such that

$$
L^\top W R = W, \tag{21}
$$

where we used the fact that the inverse of any permutation matrix is its transpose. The automorphism in Eq. (21) implies that the first term (first line) in the Hamiltonian Eq. (16) is invariant under the local non-Abelian gauge transformation. The second and third terms (second line) of Eq. (16) are invariant under the permutation of the 64 matter spins $\mu$, given that the couplings $\Gamma_\mu$ and $H_\mu$ are uniform.

*Consistency of the leg and vertex Hamiltonians*: It is important to point out that the Hamiltonians as constructed in Eqs. (11) and (16) are internally consistent. We have used two automorphisms – one on the legs in Eq. (6) and one on the vertex in Eq. (21). In fact they are consistent by construction because the vertex transformations $R$ in Eq. (18) are themselves composed of $r$ and $\ell$ matrices acting in the legs.

The transverse field terms in the Hamiltonian are not spoiled by these transformations because the monomial matrices are implemented by rotations around the $x$-axis and permutations. The fields $\Gamma_\sigma$ and $\Gamma_\nu$ are uniform and hence invariant under permutations of gauge or matter spins, respectively, on each leg (permuting spins *across* legs would not be allowed as it destroys the geometry of the lattice).

## 5 The ground state manifold

Here we establish that the ground state of the Hamiltonian Eq. (17) is an equal amplitude superposition of all the states satisfying the zero flux condition that are accessible from a reference configuration by local plaquette operations. (Ground states in distinct topological sectors are connected to different reference states.) To arrive at this result, we first consider the spectrum of a single vertex with the transverse fields switched off in Hamiltonian Eq. (16), and show in Appendix A that the lowest energy manifold of states is comprised by those respecting the zero flux condition.

To begin with, we need to ensure that even parity on each leg holds such that the quaternion representation in Eq. (1) is valid. This condition is $K > 5J/2$ applied to the couplings

between gauge and matter spins in the Hamiltonian (17). We will also require that the longitudinal field $H_\mu > 0$ in order to favor the unit flux state on each vertex. Both conditions are derived in Appendix A.

Upon turning on the transverse fields, we obtain transition matrix elements between the states satisfying the zero-flux condition in all vertices of the lattice. The lowest order terms generated in the perturbative expansion are hexagonal plaquette operators, $A_g(p)$, which multiply the six links visited by the small loop by a sequence of alternating elements $g$ and $g^{-1}$ of $Q_8$, as depicted in Fig. 1.

The amplitudes in front of each hexagonal plaquette operator $A_g$ depend on the group element $g$. The operator $A_1$ has a second order in $\Gamma$'s contribution; the operator $A_{-1}$ has the highest order in $\Gamma$'s coefficient; and all operators $A_g$, $g \neq 1, -1$, enter with equal coefficients (and at equal orders in perturbation theory). That this is the case can be seen by inspection of the basic representation of the group elements in Eq. (1). Multiplication by $g = -1$ necessarily flips all spins that represent any group element. On the other hand, multiplication by any $g \neq 1, -1$ flips exactly two spins (and similarly for $g^{-1}$). The energy of matter spins that are permuted by the operation of inserting a flux $g$ follows the same pattern because the ground state manifold is invariant under combinatorial gauge symmetry. Therefore, the effective plaquette Hamiltonian can be written as

$$H_{\text{plaquette}} = -\sum_p \sum_{g \in Q_8} \beta_g A_g(p), \tag{22}$$

where $p$ is a plaquette and $0 < \beta_{-1} < \beta_{g \neq -1, 1} < \beta_1 = 1$ (the explicit form of $\beta_g$ is not required for this argument). $H_{\text{plaquette}}$ commutes with the flux condition, as the $A_g(p)$ are the generators of the gauge symmetry.

In Kitaev's quantum double construction in Ref. [8], the weights $\beta_g$ entering the plaquette Hamiltonian are chosen to be all equal. This equal-weight choice is convenient but not required: the necessary condition is that the weights $\beta_g > 0$, to guarantee that the quantum ground state is an equal superposition of all zero-flux states (within a given topological sector). The equal amplitude superposition of such states is an eigenvector of each $A_g(p)$ and hence of $H_{\text{plaquette}}$; that this superposition is the lowest energy (ground state) eigenvector of $H_{\text{plaquette}}$ follows from the Perron-Frobenius theorem if the weights $\beta_g$ are all positive.

We thus arrive at the quantum ground state of the non-Abelian quantum double model corresponding to the quaternion group, using only 1- and 2-body interactions. In the next section we discuss a superconducting circuit with the elements needed to realize the Hamiltonian Eq. (17). If one is considering a physical system, the gap in the spin Hamiltonian would be prohibitively small given the high order of the plaquette terms. However, there is a possibility that an alternate construction may in fact have a larger (possibly non-perturbative) gap as we discuss in the next section.

# 6 Superconducting circuit realization

Instead of spins we can contemplate an array of superconducting wires connected by Josephson junctions. Such a construction would follow closely that described in Ref. [14], in which the 2-body Ising terms in the equivalent of Eq. (10) were obtained using a "waffle" where 4 vertical superconducting wires encode 4 matter degrees of freedom, and 4 horizontal wires encode 4 gauge degrees of freedom. Josephson junctions couple the horizontal and vertical wires at their 16 intersections. The ± signs that enter the matrix $w$ in Eq. (5) can in principle be realized, for instance, with regular Josephson couplings (+) or with $\pi$-junctions (−).

The larger, $W$ matrix of couplings entering in (16) and enforcing the zero-flux condition at the junctions of the honeycomb lattice would require a larger assembly, with a $64 \times 12$ mesh. The 64 wires in the construction correspond to the rows of Eq. (15), while the other 12 perpendicular wires correspond to the columns of the matrix; each element of the $64 \times 12$ $W$ matrix contains either a $+$ or a $-$ entry, which dictates whether the intersection of the vertical and horizontal wires (associated to the column and row) requires a regular Josephson coupling or a $\pi$-junction.

While the construction may appear complex, with a multitude of superconducting wires with specific types of junctions at the $64 \times 12$ crossings, one may hope that the perceived complexity may diminish if very large scale integration of superconducting circuits could ever follow the steps of semiconductor device integration. Nonetheless, complex as it may, the Hamiltonian of this system realizes *exactly* the non-Abelian gauge symmetry of the quaternion quantum double, with physical interactions.

In the limit where the superconducting wires are small, and voltage biases are tuned so two nearly degenerate charge states are favored in each wire, the system would become a non-Abelian generalization of the WXY model introduced in Ref. [14]. It is not a priori obvious whether this model would be gapped or gapless; if gapped, the scale of the gap would be non-perturbative on the only remaining energy scale in the problem, the Josephson coupling.

## 7 Conclusions

In this paper we constructed a physical spin Hamiltonian, with 1- and 2-body spin-spin interactions, which realizes a non-Abelian quantum double for the quaternion group, $Q_8$, on a honeycomb lattice. To physically represent the quaternion states, we used 4 spin-1/2 degrees of freedom. We separated the corresponding 16-dimensional Hilbert space into two subspaces of even and odd parity, and then we used the 8 even parity states to represent the 8 quaternions. We introduced additional matter spin degrees of freedom on the links and vertices of the honeycomb lattice: (i) to select the even sectors for the gauge degrees of freedom on each link; and (ii) to favor the state in which the three quaternion variables on the links connected to a lattice site multiply to the identity, i.e., to select the zero flux condition at each site as the ground state. We showed that the Hamiltonian, which includes 2-body terms that couple gauge and matter spins as well as transverse and longitudinal 1-body terms, possess an *exact* combinatorial gauge symmetry associated with the quaternion group. Notably, the spin model that we write has ferromagnetic and anti-ferromagnetic $ZZ$ interactions, plus longitudinal and transverse fields, and is thus an explicit example of a transverse Ising-type model that *does not* have a *sign* problem and yet realizes a non-Abelian topological phase. (The sign problem is absent for simulations working on the $Z$ basis, in which the $ZZ$ terms and logitudinal fields are diagonal, with the transverse fields determining the off-diagonal transition matrix elements between different configurations, always of the same sign, independent of the initial or final configuration.) We observe, however, that although the system is sign-problem-free, numerical simulation would still be difficult, as the number of spins per unit cell is large.

Our Hamiltonian would not be expected in naturally occurring materials. Instead, a physical realization of it would require a programmable quantum device. While it requires a large number of spin-1/2 degrees of freedom per unit cell to realize the system with the exact non-Abelian combinatorial gauge symmetry, the fact that it requires only 1- and 2-body spin-spin interactions makes it realistic to expect that it could be in fact programmed in a device that has the required number of qubits and the proper connectivity among them.

We also discussed a superconducting quantum circuit realization with the same non-Abelian combinatorial gauge symmetry. This type of circuit, even if it requires many elements, could conceivably be realized if large-scale integration of superconducting quantum circuits continues to advance.

## Acknowledgments

**Funding information** This work was supported by DOE Grant No. DE-FG02-06ER46316.

## A  Spectrum of a single vertex without transverse fields

Here we consider the spectrum of a single vertex with the transverse fields switched off in Hamiltonian Eq. (16), and show that that the lowest energy manifold of states is comprised by those respecting the zero flux condition.

### A.1  Even-parity spin states

Without loss of generality, we focus on the case of a vertex in sublattice A, where the three group elements $g_1, g_2$ and $g_3$ defined on the edges of the vertex multiply *clockwise*. Let $S_+$ be the set of all triplets of group elements in $Q_8$, $(g_1, g_2, g_3)$, such that $g_1 g_2 g_3 = 1$. $|S| = |Q_8|^2 = 8^2$, as choosing $g_1$ and $g_2$ fixes $g_3$. Similarly, let $S_-$ be the set of all triplets of group elements in $Q_8$, $(g_1, g_2, g_3)$, such that $g_1 g_2 g_3 = -1$. We define $S = S_+ \cup S_-$, and $\bar{S}$ the complement of $S$, i.e., all the triplets such that $g_1 g_2 g_3 \neq \pm 1$. ($|\bar{S}| = 6 \times 8^2$.)

By enumerating all the $8^3$ triplets $(g_1, g_2, g_3)$ and using their associated spin representation $[\sigma^z]$, we compute the energies given by Eq. (16) in the absence of the transverse field. Let us first compute the contributions from the 2-spin interaction terms (proportional to $J$), namely $E_J = -J [\mu^z]^\top W [\sigma^z]$. We obtain for any triplet in $S_+$ that

$$E_{S_+} = (-1J) \sum_{i=1}^{21} \mu^z_{\pi_i} + (0J) \sum_{i=22}^{45} \mu^z_{\pi_i} + (+1J) \sum_{i=46}^{63} \mu^z_{\pi_i} + (+3J) \sum_{i=64}^{64} \mu^z_{\pi_i}, \qquad (A.1)$$

for $\pi$ a permutation of the 64 indices. That is, the spectrum is independent of the state, upon permutations of the $\mu$ spins. The minimum energy is $E_{S_+}^{\min} = -42J$, attained when 21 of the $\mu^z$'s are positive, 19 are negative, and 24 can be either positive or negative (this degeneracy can be lifted with a field, see below).

Similarly, for triplets in $S_-$, the Hamiltonian reduces to

$$E_{S_-} = (+1J) \sum_{i=1}^{21} \mu^z_{\pi_i} + (0J) \sum_{i=22}^{45} \mu^z_{\pi_i} + (-1J) \sum_{i=46}^{63} \mu^z_{\pi_i} + (-3J) \sum_{i=64}^{64} \mu^z_{\pi_i}. \qquad (A.2)$$

Notice that the coefficients are opposite to those in the case of triplets in $S_+$. But the minimum energy remains at $E_{S_-}^{\min} = -42J$, attained when 21 of the $\mu^z$'s are negative, 19 are positive, and 24 can be either positive or negative.

Finally, for triplets in $\bar{S}$, the Hamiltonian reduces to

$$E_{\bar{S}} = (-2J) \sum_{i=1}^{3} \mu^z_{\pi_i} + (-1J) \sum_{i=4}^{15} \mu^z_{\pi_i} + (0J) \sum_{i=16}^{49} \mu^z_{\pi_i} + (+1J) \sum_{i=50}^{61} \mu^z_{\pi_i} + (+2J) \sum_{i=62}^{64} \mu^z_{\pi_i}, \quad (A.3)$$

with minimum energy is $E_{\bar{S}}^{\min} = -36J$.

We thus separate the manifold of states into two classes, those in $S_+$ and $S_-$ and those in $\bar{S}$ by a value $6J$. The minima for $S_+$ and $S_-$ states can be split by applying a uniform longitudinal field to the matter spins. A positive field $H_\mu$, as included in Eq. (16), lowers the energy of the $S_+$ manifold with respect to the $S_-$ one by a value $4H_\mu [= (21-19)H_\mu + (19-21)H_\mu]$.

There are degeneracies (which can be read from the number of spins multiplying the coefficient $0J$ above). These degeneracies can be lifted by the longitudinal field, or also by a transverse field.

## A.2  Inclusion of odd-parity spin states

In the above we only included the even-parity states (in terms of four spins) associated to the 8 elements of the quaternion group. Explicitly, we considered the 8 states in Eq. (1). There are the remaining 8 odd-parity states that are physical in terms of the four spins but do not correspond to quaternions. These are obtained from the "good" (even) states by, schematically,

$$\tilde{v}(\text{odd}) = v(\text{even}) \begin{bmatrix} - & 0 & 0 & 0 \\ 0 & + & 0 & 0 \\ 0 & 0 & + & 0 \\ 0 & 0 & 0 & + \end{bmatrix}. \tag{A.4}$$

Explicitly, we have the odd-parity states

$$\begin{aligned}
\tilde{v}(+1) &= \begin{bmatrix} -+++ \end{bmatrix}, & \tilde{v}(-1) &= \begin{bmatrix} +--- \end{bmatrix}, \\
\tilde{v}(+i) &= \begin{bmatrix} --+- \end{bmatrix}, & \tilde{v}(-i) &= \begin{bmatrix} ++-+ \end{bmatrix}, \\
\tilde{v}(+j) &= \begin{bmatrix} -+-- \end{bmatrix}, & \tilde{v}(-j) &= \begin{bmatrix} +-++ \end{bmatrix}, \\
\tilde{v}(+k) &= \begin{bmatrix} +++- \end{bmatrix}, & \tilde{v}(-k) &= \begin{bmatrix} ---+ \end{bmatrix}.
\end{aligned} \tag{A.5}$$

We obtain the energies of all possible states of a vertex, including legs with both positive and negative parity, by exhaustively enumerating all configurations and including the contributions from Eqs. (11) and (16) (with transverse fields off). We obtain the following energies depending on the parity of the three legs:

(i)  $E = -42J - 12K$; parities on legs: $(1,1,1)$; 128 states (64 with $g_1 g_2 g_3 = +1$ and 64 with $g_1 g_2 g_3 = -1$).

(ii)  $E = -36J - 12K$; parities on legs: $(1,1,1)$; 384 states.

(iii)  $E = -48J - 6K$; parities on legs: $(-1,-1,-1)$; 512 states.

(iv)  $E = -47J - 10K$; parities on legs: $(-1,1,1), (1,-1,1), (1,1,-1)$; 512 states each.

(v)  $E = -40J - 8K$; parities on legs: $(-1,-1,1), (-1,1,-1), (1,-1,-1)$; 512 states each.

The set of states (i), with only even-parity links, form the lowest energy manifold if the coupling constant $K$ is chosen such that $K > 5J/2$. This result, together with those in Sec. A.1 above, establish that the manifold of states corresponding to link elements $g_1, g_2, g_3 \in Q_8$ with zero flux (i.e., $g_1 g_2 g_3 = 1$) are separated from the other states by a gap $4H_\mu$.

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
