# Peer review of "Constructing Non-Abelian Quantum Spin Liquids Using Combinatorial Gauge Symmetry"

_SciPost Physics, doi:SciPost Phys. 15, 067 (2023)_

## Round 1 · Referee Report · Anonymous (Referee 1) · 2023-1-23

Report
This work introduces a spin model that realizes a quantum double with non-Abelian link representation. The novelty of the model, according to the authors, is that the model (i) has an exact gauge symmetry in all parameter range (not just emerging in low-energy limit), and (ii) contains only magnetic field terms and two-body Ising interactions, such that it is free of sign problem. Moreover, a non-Abelian topological phase is shown to appear in small transverse fields regime via perturbation theory.
Overall, I think the paper is well-written and easy to understand. Particularly, Sec. II is a nice addition to the manuscript that explains clearly the main idea behind the construction, as well as giving a brief introduction that could be helpful for non-experts. The model itself is a novel example of a model with only two-body interactions that realizes an exotic phase. As such, I recommend publication of this paper, once the following comments have been addressed.
Some comments
-As the model is free from sign problem, the authors seem to state this as one of the strong points of the model. However, the fact that the model does not have sign problem seems to not give any advantage in terms of simulatability with Quantum Monte Carlo (QMC). As a comparison, in PRB 104, 085145 (2021), the QMC simulation of a similar model only manages to simulate up to linear size L=8. This would be much worse in the current model as the number of spins per unit cell is much larger. It may be helpful to comment more on the possibility of numerical simulations of this model
- In the first sentence of Sec. IVA, it may be better to replace “triangular” with “honeycomb” to be consistent with the rest of the manuscript.
-The signs in Eq. A(1-2) appear to be incorrect. Please recheck them
-As non-zero longitudinal field is necessary to enforce the zero-flux condition, this already lifts the degeneracies for the matter spins right? The last paragraph of Appendix A1 seems to be not necessary.
Author: Dmitry Green on 2023-03-25 [id 3511]
(in reply to Report 1 on 2023-01-23)
We would like to thank referee 1 for the useful and constructive criticism. We address the referee's comments in details below.
\textit{As the model is free from sign problem, the authors seem to state this as one of the strong points of the model. However, the fact that the model does not have sign problem seems to not give any advantage n terms of simulability with Quantum Monte Carlo (QMC). As a comparison, in PRB 104, 085145 (2021), the QMC simulation of a similar model only manages to simulate up to linear size L=8. This would be much worse in the current model as the number of spins per unit cell is much larger. It may be helpful to comment more on the possibility of numerical simulations of this model}
The referee's comment is absolutely correct, while the absence of a sign problem permits simulation via Quantum Monte Carlo in principle, it does not guarantee it in practice. As the referee points out, even in our previous work on the case of Z2 combinatorial gauge symmetry we encountered such a problem, which limited the system sizes that we could study.
We have added to the text a discussion that both expands in more detail why the model does not have a sign problem, as requested by referee 2 (see below), and comment on the fact that the absence of a sign problem still does not guarantee the simulability of systems with many spins per unit cell, as pointed out by referee 1.
\textit{In the first sentence of Sec. IVA, it may be better to replace “triangular” with “honeycomb” to be consistent with the rest of the manuscript.}
Agreed; we do so in the revised manuscript.
\textit{The signs in Eq. A(1-2) appear to be incorrect. Please recheck them.}
We have checked those signs (both authors, independently), and we find that the signs are correct.
\textit{As non-zero longitudinal field is necessary to enforce the zero-flux condition, this already lifts the degeneracies for the matter spins right? The last paragraph of Appendix A1 seems to be not necessary.}
Indeed, we agree with the referee's point. We removed that paragraph.
Author: Dmitry Green on 2023-03-25 [id 3512]
(in reply to Report 2 on 2023-02-16)We thank referee 2 for the useful and constructive criticism. We address the referee's comments in details below.
The referee lists two weaknesses that we took action to address.
\textit{1- At various points when the manuscript employs certain technical terminology, the authors often cite some references. It might be helpful for a reader who is not totally familiar with this field, to have a slightly more self-contained explanations of the terminology employed.}
We have added a reference (new ref. 13) to a recent paper in which we aimed at a pedagogical introduction to combinatorial gauge theory, with a few simple examples that we work in a step-by-step manner. The objective of that paper is to generalize combinatorial gauge symmetry to any finite Abelian group. We believe that the presentation in that paper is self-contained, and we would not like to repeat material in two papers. For that reason we would like to point the reader to that reference.
In the text, the reference is now mentioned at the beginning of the third paragraph of the introduction.
\textit{2- Maybe the authors could stress a bit more in the introduction the motivation and novelty in the context of earlier recent attempts at constructions parent Hamiltonians for non-Abelian topological phases, and how in that context their current attempt stands out.}
We have inserted a discussion to that effect on a substantially expanded introduction (see a new second paragraph). In this added discussion, we address the referee's request in a direct manner, reasoning how it is that our approach effectively combines the strengths (but not the limitations) of previous efforts towardsbuilding parent Hamiltonians for non-Abelian topological phases.
The referee further lists required changes, which we implemented.
\textit{
(1) On page 2, column 1, last para: Could the authors please elaborate on what they mean by “while the matter spins are introduced so as to enable the symmetries.”?}
We have added a sentence at the pertinent paragraph to specify that the symmetry is enabled via permutations of states within the enlarged Hilbert space.
\textit{
(2) In Sec. IV, first para, is there a simple way of seeing why “the other set of 8 states – those with odd parity – will be pushed up in energy.”?}
We are not able (unfortunately) to offer a simple way to see this result; Sec. IV A is needed to establish it, as well as the results of Ref. 12. To help the reader identify where the stated results are derived, we added to the two introductory paragraphs of Sec. IV the references to the appropriate subsections where the results are obtained.
\textit{
(3) On Page 6, column 1, para 1, the authors claim that the commutativity of the R matrices ensures that one can insert charges locally. It is clear that this is a necessary condition, however, could the authors clarify and maybe brieOy explain in the manuscript how this condition turns out to be sufficient to ensure that one can locally insert charges.}
We clarified in the revised text that the two operations of inserting a group element into one hexagon and another group element into a neighboring hexagon correspond to multiplication on the left and right, which commute.
\textit{
(4) It would be helpful to add a sentence or two which explicitly explains why there is no sign problem in the obtained Hamiltonian.}
We have expanded the discussion of the absence of a sign problem (as suggested by both referees).
\textit{
(5) The l(h) and r(h) matrices are defined differently than the “standard” representation adopted like in Ref. [14]. Could the authors briefly mention in the manuscript the reason/advantage for adopting the slightly different choice of matrices.}
We now explain in the text, before Eq. (1) is presented, that we introduce and use different matrices that utilize elements ±1 (rather than 0's and 1's) as entries for the vectors representing the elements of Q8. We make this choice as we have in mind that each entry corresponds to the z-component of a spin. (The representation with the 1's and 0's could be viewed, alternatively, as the presence or absence of a charge.)

---

## Round 1 · Referee Report · Anonymous (Referee 2) · 2023-2-16

Strengths
1- The authors address a much pursued problem of constructing a model Hamiltonian which realizes a non-Abelian topological phase as its ground state.
2- Remarkably, this Hamiltonian features only 1- and 2-body interactions of the Ising type with the important addition of longitudinal and transverse magnetic field terms, the latter proving crucial in stabilizing the non-Abelian phase as demonstrated within a perturbative treatment.
3- As opposed to "conventional" slave-particle constructions where there is an emergent low-energy gauge theory, the authors here reveal the interesting property of the presence of a universal gauge symmetry.
4- The model the construct is sign problem free.
Weaknesses
1- At various points when the manuscript employs certain technical terminology, the authors often cite some references. It might be helpful for a reader who is not totally familiar with this field, to have a slightly more self-contained explanations of the terminology employed.
2- Maybe the authors could stress a bit more in the introduction the motivation and novelty in the context of earlier recent attempts at constructions parent Hamiltonians for non-Abelian topological phases, and how in that context their current attempt stands out.
Report
The manuscript is well written, follows a deductive logic, and the arguments are scientifically sound and conveyed well albeit at some points quite tersely. I think this is an important contribution to the field and a solid work which deserves to be published. The authors may consider incorporating some of the recommendations provided, and I would be happy to hear their responses even if brief.
Requested changes
Some recommendations/suggestions:
(1) On page 2, column 1, last para: Could the authors please elaborate on what they mean by “while the matter spins are introduced so as to enable the symmetries.”?
(2) In Sec. IV, first para, is there a simple way of seeing why “the other set of 8 states – those with odd parity – will be pushed up in energy.”?
(3) On Page 6, column 1, para 1, the authors claim that the commutativity of the R matrices ensures that one can insert charges locally. It is clear that this is a necessary condition, however, could the authors clarify and maybe briefly explain in the manuscript how this condition turns out to be sufficient to ensure that one can locally insert charges.
(4) It would be helpful to add a sentence or two which explicitly explains why there is no sign problem in the obtained Hamiltonian.
(5) The l(h) and r(h) matrices are defined differently than the “standard” representation adopted like in Ref. [14]. Could the authors briefly mention in the manuscript the reason/advantage for adopting the slightly different choice of matrices.

---

## Round 2 · Referee Report · Anonymous (Referee 2) · 2023-4-4

Report

The authors have responded to all questions of the referees in a convincing way, and incorporated all changes in the revised manuscript in an appropriate manner. I therefore suggest the manuscript be published in this form.

---

## Round 2 · Referee Report · Anonymous (Referee 1) · 2023-4-14

Report

The authors have addressed all of the referees' comments appropriately. I now recommend publication of this manuscript as is.

---

## Round 2 · Author Response

We thank both referees' time to review and comment on our paper, and
for their constructive criticisms. We have incorporated their comments
and suggestions into our revised manuscript.

---

## Round 2 · List of Changes

Reply to Referee 1

We would like to thank referee 1 for the useful and constructive criticism. We address the referee's comments in details below.

\textit{As the model is free from sign problem, the authors seem to state this as one of the strong points of the model. However, the fact that the model does not have sign problem seems to not give any advantage n terms of simulability with Quantum Monte Carlo (QMC). As a comparison, in PRB 104, 085145 (2021), the QMC simulation of a similar model only manages to simulate up to linear size L=8. This would be much worse in the current model as the number of spins per unit cell is much larger. It may be helpful to comment more on the possibility of numerical simulations of this model}

The referee's comment is absolutely correct, while the absence of a sign problem permits simulation via Quantum Monte Carlo in principle, it does not guarantee it in practice. As the referee points out, even in our previous work on the case of $\mathbb{Z}_2$ combinatorial gauge symmetry we encountered such a problem, which limited the system sizes that we could study.

We have added to the text a discussion that both expands in more detail why the model does not have a sign problem, as requested by referee 2 (see below), and comment on the fact that the absence of a sign problem still does not guarantee the simulability of systems with many spins per unit cell, as pointed out by referee 1.

\textit{In the first sentence of Sec. IVA, it may be better to replace “triangular” with “honeycomb” to be consistent with the rest of the manuscript.}

Agreed; we do so in the revised manuscript.

\textit{The signs in Eq. A(1-2) appear to be incorrect. Please recheck them.}

We have checked those signs (both authors, independently), and we find that the signs are correct.

\textit{As non-zero longitudinal field is necessary to enforce the zero-flux condition, this already lifts the degeneracies for the matter spins right? The last paragraph of Appendix A1 seems to be not necessary.}

Indeed, we agree with the referee's point. We removed that paragraph.

Reply to Referee 2

We also thank referee 2 for the useful and constructive criticism. We address the referee's comments in details below.

The referee lists two weaknesses that we took action to address.

\textit{1- At various points when the manuscript employs certain technical terminology, the authors often cite some references. It might be helpful for a reader who is not totally familiar with this field, to have a slightly more self-contained explanations of the terminology employed.}

We have added a reference (new ref. 13) to a recent paper in which we aimed at a pedagogical introduction to combinatorial gauge theory, with a few simple examples that we work in a step-by-step manner. The objective of that paper is to generalize combinatorial gauge symmetry to any finite Abelian group. We believe that the presentation in that paper is self-contained, and we would not like to repeat material in two papers. For that reason we would like to point the reader to that reference.

In the text, the reference is now mentioned at the beginning of the third paragraph of the introduction.

\textit{2- Maybe the authors could stress a bit more in the introduction the motivation and novelty in the context of earlier recent attempts at constructions parent Hamiltonians for non-Abelian topological phases, and how in that context their current attempt stands out.}

We have inserted a discussion to that effect on a substantially expanded introduction (see a new second paragraph). In this added discussion, we address the referee's request in a direct manner, reasoning how it is that our approach effectively combines the strengths (but not the limitations) of previous efforts towardsbuilding parent Hamiltonians for non-Abelian topological phases.

The referee further lists required changes, which we implemented.

\textit{
(1) On page 2, column 1, last para: Could the authors please elaborate on what they mean by “while the matter spins are introduced so as to enable the symmetries.”?}

We have added a sentence at the pertinent paragraph to specify that the symmetry is enabled via permutations of states within the enlarged Hilbert space.

\textit{
(2) In Sec. IV, first para, is there a simple way of seeing why “the other set of 8 states – those with odd parity – will be pushed up in energy.”?}

We are not able (unfortunately) to offer a simple way to see this result; Sec. IV A is needed to establish it, as well as the results of Ref. 12. To help the reader identify where the stated results are derived, we added to the two introductory paragraphs of Sec. IV the references to the appropriate subsections where the results are obtained.

\textit{
(3) On Page 6, column 1, para 1, the authors claim that the commutativity of the R matrices ensures that one can insert charges locally. It is clear that this is a necessary condition, however, could the authors clarify and maybe brieOy explain in the manuscript how this condition turns out to be sufficient to ensure that one can locally insert charges.}

We clarified in the revised text that the two operations of inserting a group element into one hexagon and another group element into a neighboring hexagon correspond to multiplication on the left and right, which commute.

\textit{
(4) It would be helpful to add a sentence or two which explicitly explains why there is no sign problem in the obtained Hamiltonian.}

We have expanded the discussion of the absence of a sign problem (as suggested by both referees).

\textit{
(5) The l(h) and r(h) matrices are defined differently than the “standard” representation adopted like in Ref. [14]. Could the authors briefly mention in the manuscript the reason/advantage for adopting the slightly different choice of matrices.}

We now explain in the text, before Eq. (1) is presented, that we introduce and use different matrices that utilize elements $\pm 1$ (rather than 0's and 1's) as entries for the vectors representing the elements of $Q_8$. We make this choice as we have in mind that each entry corresponds to the z-component of a spin. (The representation with the 1's and 0's could be viewed, alternatively, as the presence or absence of a charge.)

---

## Editorial Decision

published